# Development of an Immunochromatography Assay to Detect Marburg Virus and Ravn Virus

**DOI:** 10.3390/v15122349

**Published:** 2023-11-29

**Authors:** Katendi Changula, Masahiro Kajihara, Shino Muramatsu, Koji Hiraoka, Toru Yamaguchi, Yoko Yago, Daisuke Kato, Hiroko Miyamoto, Akina Mori-Kajihara, Asako Shigeno, Reiko Yoshida, Corey W. Henderson, Andrea Marzi, Ayato Takada

**Affiliations:** 1Department of Paraclinical Studies, School of Veterinary Medicine, University of Zambia, Lusaka 10101, Zambia; katendi.changula@sacids.org; 2Division of Global Epidemiology, International Institute for Zoonosis Control, Hokkaido University, Sapporo 001-0020, Japan; kajihara@czc.hokudai.ac.jp (M.K.); hirom@czc.hokudai.ac.jp (H.M.); akinam@czc.hokudai.ac.jp (A.M.-K.); shigeno-a@czc.hokudai.ac.jp (A.S.); yoreiko@hotmail.com (R.Y.); 3DENKA Co., Ltd., Tokyo 103-8338, Japan; shino-muramatsu@denka.co.jp (S.M.); koji-hiraoka@denka.co.jp (K.H.); toru-yamaguchi@denka.co.jp (T.Y.); yoko-yago@denka.co.jp (Y.Y.); daisuke-kato@denka.co.jp (D.K.); 4Laboratory of Virology, Division of Intramural Research, National Institute of Allergy and Infectious Diseases, National Institutes of Health, Hamilton, MT 59840, USA; 5International Collaboration Unit, International Institute for Zoonosis Control, Hokkaido University, Sapporo 001-0020, Japan; 6One Health Research Center, Hokkaido University, Sapporo 001-0020, Japan; 7Department of Disease Control, School of Veterinary Medicine, University of Zambia, Lusaka 10101, Zambia

**Keywords:** Marburg virus, MARV, Ravn virus, filovirus, nucleoprotein, monoclonal antibody, immunochromatography assay, diagnosis, rapid diagnostic test, RDT

## Abstract

The recent outbreaks of Marburg virus disease (MVD) in Guinea, Ghana, Equatorial Guinea, and Tanzania, none of which had reported previous outbreaks, imply increasing risks of spillover of the causative viruses, Marburg virus (MARV) and Ravn virus (RAVV), from their natural host animals. These outbreaks have emphasized the need for the development of rapid diagnostic tests for this disease. Using monoclonal antibodies specific to the viral nucleoprotein, we developed an immunochromatography (IC) assay for the rapid diagnosis of MVD. The IC assay was found to be capable of detecting approximately 10^2−4^ 50% tissue culture infectious dose (TCID_50_)/test of MARV and RAVV in the infected culture supernatants. We further confirmed that the IC assay could detect the MARV and RAVV antigens in the serum samples from experimentally infected nonhuman primates. These results indicate that the IC assay to detect MARV can be a useful tool for the rapid point-of-care diagnosis of MVD.

## 1. Introduction

Marburg virus disease (MVD) and Ebola virus disease (EVD), caused by viruses of the family *Filoviridae*, are acute febrile diseases often manifested as hemorrhagic fevers of human and nonhuman primates [1]. MVD is caused by Marburg virus (MARV) and Ravn virus (RAVV), both belonging to the species *Orthomarburgvirus marburgense* in the genus *Orthomarburgvirus*, while EVD is caused by five viruses of the genus *Orthoebolavirus*: Ebola virus (EBOV), Sudan virus (SUDV), Taï Forest virus (TAFV), Bundibugyo virus (BDBV), and Reston virus (RESTV), representing the species *Orthoebolavirus zairense*, *Orthoebolavirus sudanense*, *Orthoebolavirus taiense*, *Orthoebolavirus bundibugyoense*, and *Orthoebolavirus restonense*, respectively [2,3].

The incidence of MVD in Africa has been increasing [4,5]. From the first outbreak of MVD in Germany and Yugoslavia in 1967, which originated from African green monkeys imported from Uganda, a total of 17 MVD outbreaks in humans have been recorded, with the most recent being reported from Equatorial Guinea and Tanzania in 2023 [5,6]. The largest ever recorded outbreak of MVD occurred in Angola in 2005, with a case fatality rate of 90% [5]. Diagnosis is complicated by the fact that the initial symptoms of MVD, such as weakness, chills, headache, fever, and arthralgia, are nonspecific, followed by diarrhea, nausea, vomiting, and in some cases rash and hemorrhage [7,8]. A further challenge in MVD diagnosis is that these outbreaks occur in areas that are remote and lack adequate healthcare infrastructure and personnel to diagnose the disease, resulting in an increased potential spread of the infection [9,10,11]. Therefore, there is a need for simple, rapid, and virus-specific diagnostic tests that can be deployed in such areas to aid in the reduction in transmission.

Orthomarburgvirus particles consist of at least seven structural proteins: nucleoprotein (NP), viral protein (VP) 35, VP40, glycoprotein (GP), VP30, VP24, and RNA-dependent RNA polymerase [12]. Of these, NP appears to be an ideal target protein for antigen detection assays since it is abundant in viral particles, has strong antigenicity, and is generally expected to have common epitopes among MARV and RAVV variants [13,14,15]. Using monoclonal antibodies (mAbs) to EBOV NP, we previously developed a rapid diagnostic test (RDT) for detection of the orthoebolaviruses EBOV, TAFV, and BDBV in an immunochromatography (IC) assay [16,17,18]. In this paper, we report the development of a similar IC assay for the detection of MARV and RAVV using NP-specific mAbs.

## 2. Materials and Methods

### 2.1. Viruses and Cells

MARV (Musoke, Angola, Ozolin, and Ci67), RAVV (Ravn), EBOV (Mayinga 76), SUDV (Boniface), BDBV (Butalya), RESTV (Pennsylvania), Lassa virus (Josiah), and Crimean–Congo hemorrhagic fever virus (Hoti) were propagated in African green monkey kidney Vero E6 cells and stored at −80 °C. Virus titers were determined as the 50% tissue culture infectious dose (TCID_50_) using Vero E6 cells. All experiments involving the use of infectious viruses were performed in the biosafety level (BSL) 4 laboratories of the Integrated Research Facility at the Rocky Mountain Laboratories, Division of Intramural Research, National Institute of Allergy and Infectious Diseases, National Institutes of Health (Hamilton, MT, USA). Standard operating procedures for infectious work were approved by the Rocky Mountain Laboratories Biosafety Committee.

### 2.2. Preparation of Virus-like Particles (VLPs) and Purified Recombinant NPs (rNPs)

Plasmids encoding NP, VP40, and GP of MARV (Musoke) were constructed as described previously [16]. VLPs were produced by the transfection of human embryonic kidney 293T cells with plasmids expressing MARV NP, VP40, and GP as described previously [19]. Forty-eight hours after transfection, supernatants containing VLPs were harvested, purified, and used for the immunization of mice and the initial evaluation of lateral-flow IC assays. For the preparation of purified rNPs, 293T cells were transfected with the plasmids expressing rNPs. Forty-eight hours later, the cells were lysed, and the rNP fraction was collected by discontinuous CsCl gradient centrifugation as described previously [20,21]. The rNPs were used as antigens for enzyme-linked immunosorbent assay (ELISA) and Western blotting as described previously [16].

### 2.3. Mouse mAbs for the Preparation of IC Assay Devices

mAbs were generated as described previously [16]. Briefly, six-week-old female Balb/c mice were immunized twice intramuscularly with VLPs with complete or incomplete Freund’s adjuvant (Difco). Then, the animals were intravenously boosted with the same VLPs without adjuvant. Spleen cells were harvested on day 4 after the booster injection and fused to myeloma cells. Hybridoma supernatants were screened by ELISA for the secretion of NP-specific antibodies, using purified rNP as antigens. Selected hybridoma cells were then cloned twice with limiting dilution methods. The animal protocol was approved by the Animal Care and Use Committee of Hokkaido University on 30 March 2018 (#18-0029). Using the selected mAbs, IC assay devices were produced as described previously [17]. For each assay, 20–25 μL of supernatants, VLPs, and serum samples were used. Some control experiments involving the use of human sera were approved by the institutional ethics committee, (Denka, Niigata, Japan) in accordance with the Declaration of Helsinki.

### 2.4. Nonhuman Primate (NHP) Serum Samples

NHP serum samples (from rhesus and cynomolgus macaques) collected during previous studies were used [22,23,24]. The collection of the samples complied with the Animal Welfare Act and other federal statutes and regulations relating to animals and experiments involving animals and adhered to the principles stated in the Guide for the Care and Use of Laboratory Animals (National Research Council, 2011). The samples were stored at −80 °C until use. Gamma-irradiated serum samples were used for the evaluation in BSL-2 laboratories.

## 3. Results

### 3.1. Selection of mAbs for the IC Assay

We generated 13 clones of NP-specific mAbs and analyzed their binding capacities to the rNPs of the representative isolates of MARV (Angola and Musoke) and RAVV (Ravn) as well as EBOV, SUDV, TAFV, BDBV, and RESTV in ELISA and/or Western blotting (Table 1 and Appendix A). We then used 11 mAbs that reacted to both MARV and RAVV in order to select two mAbs suitable for the IC assay (i.e., labeled and capture mAbs). We produced tentative test devices based on a lateral flow IC assay using purified mAbs for all combinations of these 11 mAbs as labeled and capture mAbs. These devices were evaluated for their ability to detect rNPs using normal human serum mixed with appropriately diluted MARV VLPs. Band intensities of the test line were visually scored for the sensitivity of IC assays. Nonspecific reactions caused by serum components were also monitored by concurrently testing normal human serum without adding VLPs. Then, we selected the best combination of mAbs that offered the highest sensitivity and lowest nonspecific reaction. Using the selected mAbs, MNP 6H9-5.1-6 and MNP 1D9-9-1, as labeled and capture mAbs, respectively, we produced IC devices for further evaluations as described previously for the QuickNavi^TM^-Ebola device [17]. As with the QuickNavi^TM^-Ebola, blood or serum samples (10–30 μL) can be directly applied onto the sample pad of this IC assay device, followed by the addition of two drops (approximately 40 μL) of the sample buffer (saline-based reagent) supplied together with the test kit. The results can be interpreted 10–20 min later as positive by the appearance of both control and test lines or as negative if only the control line appears (Figure 1).

### 3.2. Sensitivity and Specificity of the IC Assay to Detect MARV and RAVV in Tissue Culture Supernatants

We first assessed the specificity and sensitivity of the IC assay using tissue culture supernatants from Vero E6 cells infected with MARV, RAVV, EBOV, SUDV, BDBV, or RESTV. The supernatants (10^5^–10^7^ TCID_50_/mL) were 10-fold serially diluted and each dilution was applied onto the IC device (Table 2). We found that the assay was reactive for MARV and RAVV but not for EBOV, SUDV, BDBV, or RESTV, even at the highest concentrations of the viruses. The limit of detection (LOD) was 2 × 10^2−4^ TCID_50_/test (i.e., in 20 μL) depending on the virus. Lassa virus (5.6 × 10^6^ TCID_50_/mL) and Crimean–Congo hemorrhagic fever virus (1.7 × 10^6^ TCID_50_/mL) were not detected by the assay.

### 3.3. Performance of the IC Assay in NHP Models of MVD

We next evaluated the utility of the IC assay, using serum samples collected from five, three, and three NHPs experimentally infected with MARV (Angola), RAVV, and EBOV, respectively (Table 3). Virus titers in these samples had already been determined in the previous studies [22,23,24]. Serum samples were collected on days 0, 3, 6, and 7–9 after infection. Undiluted serum samples were directly applied to the sample pad of the IC device, and the results were obtained 10–20 min later. We found that MARV and RAVV NP antigens were detected in most of the samples containing infectious virus particles that were detectable in TCID_50_ assays. Consistent with the data of the infectious tissue culture experiment, the IC assay seemed to be able to detect approximately 10^2−4^ TCID_50_ of MARV and RAVV in the applied serum samples (25 μL). As expected, EBOV was not detected in the infected NHPs even at the terminal stage of the infection (i.e., day 6 or later). To further confirm the LOD of the IC assay, 2-fold dilutions of the MARV (Angola) supernatant were artificially mixed with negative control NHP serum and tested using the IC assay (Table 4). The assay detected 2.5 × 10^4^ but not 1.25 × 10^4^ TCID_50_/mL of the virus, suggesting that the LOD was 5.0 × 10^2^ TCID_50_/test for Angola (i.e., 20 μL), which was similar to that shown in Table 2.

## 4. Discussion

MVD is a zoonosis endemic to Africa [25]. The reservoir host has been determined to be Egyptian fruit bats (*Rousettus aegyptiacus*), and transmission to humans has been linked to entry into caves/mines where these bats roost [7,26,27,28,29]. Since 2021, there have been reports of MVD outbreaks every year to date, all from countries that were not known to be endemic to MARV or RAVV [5]. In addition, some countries with no recorded MVD outbreak, such as Gabon, Zambia, Sierra Leone, and South Africa, have reported detection of MARV and/or RAVV in Egyptian fruit bats [30,31,32,33]. The increased incidence of MVD, particularly in areas with no previously reported outbreak, as well as the detection of the virus in the bats in nonendemic countries, underscores that MARV and RAVV infections are emerging and re-emerging zoonoses that pose threats to human health [4,11]. The World Health Organization has designated MVD as a priority pathogen for research and development due to the risk to public health [34]. However, despite being the first member of the family *Filoviridae* to be discovered, research into diagnostics and therapeutics for MVD has lagged behind that for EVD [11].

While nucleic acid detection assays are currently used for the early detection of MARV infection, they require specialized laboratories and personnel. Antibody detection serological tests are not ideal for early diagnosis as IgM antibodies can only be detected several days after the onset of symptoms, while IgG antibodies can be detected in the second week of infection and persist for years [35]. In most MVD outbreaks, rapid diagnosis is hampered by the lack of laboratory infrastructure, resources, and trained personnel as well as the remoteness of areas that normally experience outbreaks and the lack of specific clinical symptoms, resulting in large outbreaks [9,10,11]. Therefore, there is a need for diagnostic tests, such as RDTs, that can detect MARV and RAVV antigens early during infection. The most commonly used RDTs are single-use IC assays that rely on antibodies to detect viral antigens in a clinical sample [36]. In general, RDTs are quite useful as they are portable, easy to use, with no specialized equipment or training required, have a quick turn-around time, can be stored at room temperature for extended periods, are relatively cheap, and can be employed at the point of care settings [35,36,37]. However, there are currently no RDTs available for the detection of MARV and RAVV. Thus, we developed the IC assay to detect both MARV and RAVV antigens using mAbs specific to NP.

NP is an ideal target for detection because of its abundance since each virion has approximately 3170 copies of NP [13]. The N-terminal of the NP is highly conserved, while the C-terminal is variable and highly antigenic [13,14,38]. Of the 13 anti-NP mAbs generated in this study, 11 showed cross-reactivity to both MARV and RAVV. This is not unusual considering that there is only a 16% genetic difference between MARV and RAVV at the nucleotide level [39]. Additionally, it appears that anti-NP mAbs have a tendency to be cross-reactive within the filovirus genera. In our previous studies, of the 10 mAbs to EBOV NP that were generated, 4 were cross-reactive to SUDV, TAFV, BDBV, and RESTV and 4 were cross-reactive to at least one of the other orthoebolaviruses mentioned above [16,17]. mAbs that are cross-reactive to multiple viruses within the same genus are ideal to use for such assays as they can potentially identify even novel viral species. Indeed, the diagnosis during the first outbreak caused by BDBV was made using an antigen detection ELISA with mAbs cross-reactive to EBOV, SUDV, and RESTV following an initial negative RT-PCR test [40]. The mAbs produced in this study were reactive to MARV and RAVV but not to any orthoebolaviruses tested. Our previous studies also showed that there was no cross-reactivity of EBOV NP mAbs to MARV [16,17]. Thus, it might be difficult to produce highly cross-reactive mAbs to both orthoebolavirus and orthomarburgvirus, suggesting the use of antibody cocktails for developing pan-filovirus RDTs.

The LOD of our IC assay for infectious tissue culture supernatants was 2 × 10^2−4^ TCID_50_/test for MARV and RAVV. The assay performance with infected NHP sera was generally consistent with this LOD. While there is a lack of detailed information on the onset of detectable viremia in MARV- and RAVV-infected humans, studies on experimentally infected NHPs have reported that viremia occurs 3 to 6 days post-infection, coinciding with the onset of fever [41,42,43,44]. Our IC assay did not detect the viruses on day 3, but was able to detect them on day 6 for all the infected NHPs. Since the NHP sera was only collected at 3-day intervals until death or recovery, we could not definitively determine that day 6 was the onset of viremia, or whether the IC assay would be able to detect viremia that might occur on days 4 or 5. Since the virus titers were uniformly quite high on day 6, we assume that viremia might start before day 6. Regardless, we infer from the LOD for infectious tissue culture supernatants that the IC assay can detect the virus shortly after the onset of viremia.

We previously developed an IC-based RDT, QuickNavi^TM^-Ebola, using mAbs to EBOV NP that detects EBOV, TAFV, and BDBV and can be stored at room temperature for at least 24 months [17,18]. It was deployed during the 2018–2022 outbreak of EVD in the Democratic Republic of the Congo with a reported sensitivity and specificity of 85% and 99.8% [18] and 87.4% and 99.6%, respectively [45]. Indeed, QuickNavi^TM^-Ebola showed comparable performance to other WHO-approved RDTs [18,45,46]. One limitation of the present study is that a positive IC assay test could not be related to the onset of viremia due to the 3-day intervals between the collection of the infected NHP sera, unlike our previous study on QuickNavi^TM^-Ebola where the collection was at shorter intervals and the majority of NHPs tested positive upon the onset of viremia [17]. Another limitation is that there was a limited number of NHP infectious samples tested and that the IC assay was not tested for known positive and negative human samples. Although our IC device for MARV detection needs to be clinically tested in the future, it is expected to be a useful tool for the initial diagnosis and point-of-care screening of MVD, particularly in areas that are remote and lack appropriate diagnostic infrastructure to limit further transmission by the isolation of suspected individuals until a confirmatory diagnosis can be made.

## Figures and Tables

**Figure 1 viruses-15-02349-f001:**
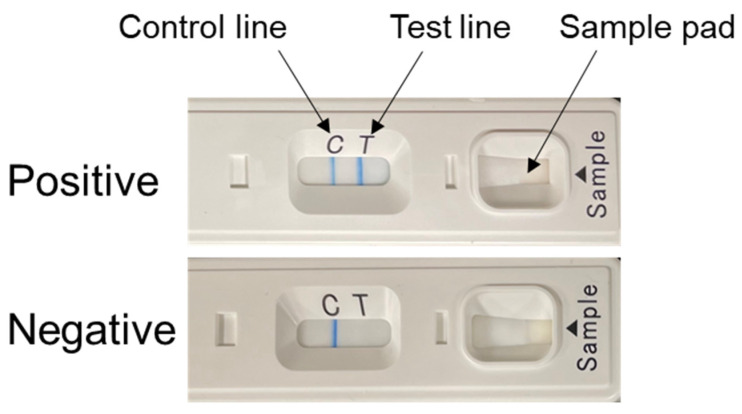
Appearance of the device and interpretation of the results of the IC assay.

**Table 1 viruses-15-02349-t001:** Binding profiles of the anti-NP mAbs produced in this study.

mAb Clone	Ig Class	Reactivity ^a^			
		Musoke	Angola	Ravn	Orthoebolaviruses ^b^
MNP 13-10-1	IgG1	+	+	+	-
MNP 31-8-1	IgG1	+	-	-	-
MNP 52-2-2	IgG2b	+	+	+	-
MNP 95-4-2-6	IgG2b	+	+	+	-
MNP 98-3-8	IgG1	+	+	+	-
MNP 121-9-5	IgG1	+	+	+	-
MNP 6H9-5.1-6 ^c^	IgG1	+	+	+	-
MNP 1D9-9-1 ^c^	IgG3	+	+	+	-
MNP 1G5-1-5	IgG1	+	+	+	-
MNP 4H6-7-1	IgG3	+	+	+	-
MNP 6F1-7-1-7	IgM	+	+	+	-
MNP 8A3-3-1-4-2-1	IgG1	+	+	-	-
MNP 15F8-2-1	IgG1	+	+	+	-

^a^ Reactivity was confirmed by ELISA and/or Western blotting using rNPs. ^b^ EBOV, SUDV, BDBV, TAFV, and RESTV rNPs were used. ^c^ mAbs selected for the IC device for the following tests.

**Table 2 viruses-15-02349-t002:** Detection of infectious filoviruses by the IC device.

Genus	Virus	Isolates	Titer (TCID_50_/mL)	Dilution (10^n^)				LOD/20 μL
				0	1	2	3	
*Orthomarburgvirus*	MARV	Musoke	5.0 × 10^5^	++	+	-	-	1.0 × 10^3^
	MARV	Angola	1.0 × 10^6^	++	++	+	-	2.0 × 10^2^
	MARV	Ozolin	7.0 × 10^6^	++	+	-	-	1.4 × 10^4^
	MARV	Ci67	1.0 × 10^7^	++	++	-	-	2.0 × 10^4^
	RAVV	Ravn	1.0 × 10^6^	++	+	-	-	2.0 × 10^3^
*Orthoebolavirus*	EBOV	Mayinga 76	1.0 × 10^6^	-	-	-	-	NA
	SUDV	Boniface	3.0 × 10^5^	-	-	-	-	NA
	BDBV	Butalya	1.0 × 10^5^	-	-	-	-	NA
	RESTV	Pennsylvania	7.0 × 10^5^	-	-	-	-	NA

++: Strongly positive, +: Positive, -: Negative, NA: Not applicable.

**Table 3 viruses-15-02349-t003:** Detection of Marburg and Ravn viruses in the sera of infected NHPs by the IC device.

NHP ID	Day after Infection	IC Assay Result	TCID_50_/mL Blood
MARV#1 (rhesus)	0	Not detected	Not detected
	3	Not detected	Not detected
	6	Detected	3.2 × 10^8^
	7 ^a^	Detected	5.6 × 10^7^
MARV#2 (rhesus)	0	Not detected	Not detected
	3	Not detected	Not detected
	6	Detected	3.2 × 10^7^
	8 ^a^	Detected	3.2 × 10^6^
MARV#3 (cynomolgus)	0	Not detected	Not detected
	3	Not detected	Not detected
	6	Detected	5.6 × 10^7^
	7 ^a^	Detected	1.8 × 10^8^
MARV#4 (cynomolgus)	0	Not detected	Not detected
	3	Not detected	Not detected
	6	Detected	5.6 × 10^7^
	7 ^a^	Detected	1.8 × 10^8^
MARV#5 (cynomolgus)	0	Not detected	Not detected
	3	Not detected	1.6 × 10^4^
	6	Detected	3.4 × 10^7^
	7 ^a^	Not determined	Not determined
RAVV#1 ^b^ (rhesus)	0	Not detected	Not detected
	3	Not detected	Not detected
	6	Detected ^c^	3.2 × 10^3^
	9	Detected	Not detected
	12 ^b^	Not detected	Not detected
RAVV#3 (cynomolgus)	0	Not detected	Not detected
	3	Not detected	Not detected
	6	Detected	3.2 × 10^5^
	9 ^a^	Detected	1.8 × 10^7^
RAVV#4 (cynomolgus)	0	Not detected	Not detected
	3	Not detected	Not detected
	6	Detected	3.2 × 10^5^
	9 ^a^	Detected	3.2 × 10^8^
EBOV#1 (cynomolgus)	0	Not detected	Not detected
	3	Not detected	3.2 × 10^3^
	5 ^a^	Not detected	1.8 × 10^8^
EBOV#2 (cynomolgus)	0	Not detected	Not detected
	3	Not detected	Not detected
	6	Not detected	3.2 × 10^6^
	7 ^a^	Not detected	1.8 × 10^6^
EBOV#3 (cynomolgus)	0	Not detected	Not detected
	3	Not detected	Not detected
	6	Not detected	3.2 × 10^7^
	7 ^a^	Not detected	1.8 × 10^8^

^a^ Terminal, died; ^b^ Survived; ^c^ Only weakly detected.

**Table 4 viruses-15-02349-t004:** Detection of MARV spiked in NHP ^a^ serum.

Isolate	Titer (TCID_50_/mL)					LOD/20 μL
	1.0 × 10^5^	5.0 × 10^4^	2.5 × 10^4^	1.3 × 10^4^	6.1 × 10^3^	
Angola	++	++	+	-	-	5.0 × 10^2^

^a^ Uninfected cynomolgus macaque. ++: Strongly positive, +: Positive, -: Negative

## Data Availability

All relevant data are provided in the manuscript.

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
