# Peer review of "Development of an Immunochromatography Assay to Detect Marburg Virus and Ravn Virus"

_viruses, 2023, doi:10.3390/v15122349_

Round 1

Reviewer 1 Report

Comments and Suggestions for Authors

Changula et al. have developed an immunochromatographic assay for the detection of the Marburg virus and Ravn virus by using NP-specific mAbs. The reviewer has the following comments.

- In the abstract for 50% tissue culture infectious dose do also mention TCID50 in the bracket for ease of readability.

- Explain a bit more about the methodology and interpretation for the 'binding profile' as well as how the 'highest sensitivity and lowest non-specific result' was determined.

- In Table 1, suggest writing orthoebolavirus instead of Ebola.

- Reactivity determined by ELISA and /or Western-blot data can be provided as a supplementary figure.

- Discuss the sensitivity of the assay as it failed to detect TCID50/mL blood of 1.6 x 104 in MARV#5. Also in most of the monkeys, the detection was made after 6 days. Considering early detection would be beneficial, the authors can discuss how the assay fairs to any antibody detection assay (IgM or IgG) or Nucleic acid-based detection if performed and emphasize early and late-stage detection.

- How does this assay compare to any similar assays available? 

- The LOD is determined based on the cell culture supernatant. Could the authors have used any alternate methods alongside? Also, any spiking in serum and whole blood samples with virus supernatant have been tried?

- Mention the limitations of the study in the discussion.

Author Response

Reviewer #1

- In the abstract for 50% tissue culture infectious dose do also mention TCID50 in the bracket for ease of readability.

Response: Thank you for the suggestion. We added “TCID50” in the bracket.

- Explain a bit more about the methodology and interpretation for the 'binding profile' as well as how the 'highest sensitivity and lowest non-specific result' was determined.

Response: Thank you for the suggestion. We modified this paragraph as follows: “We generated 13 clones of NP-specific mAbs and analyzed their binding capacities to rNPs of the representative isolates of MARV (Angola and Musoke) and RAVV (Ravn) as well as EBOV, SUDV, TAFV, BDBV, and RESTV in ELISA and/or western blotting (Table 1 and Figure S1). We then used 11 mAbs that reacted to both MARV and RAVV in order to select two mAbs suitable for the IC assay (i.e., labeled and capture mAbs). We produced tentative test devices based on a lateral-flow IC assay using purified mAbs for all combinations of these 11 mAbs as labeled and capture mAbs. These devices were evaluated for their ability to detect rNPs using normal human serum mixed with ap-propriately diluted MARV VLPs. Band intensities of the test line were visually scored for the sensitivity of IC assays. Nonspecific reactions caused by serum components were also monitored by concurrently testing normal human serum without adding VLPs. Then we selected the best combination of mAbs that gave the highest sensitivity and lowest nonspecific reaction” (Lines 117-129).

- In Table 1, suggest writing orthoebolavirus instead of Ebola.

Response: Thank you for the suggestion. We replaced Ebola with orthoebolavirus.

- Reactivity determined by ELISA and/or Western-blot data can be provided as a supplementary figure.

Response: Thank you for the suggestion. We provided Western blot images as a Supplementary Figure (Figure S1).

- Discuss the sensitivity of the assay as it failed to detect TCID50/mL blood of 1.6 x 104 in MARV#5. Also in most of the monkeys, the detection was made after 6 days. Considering early detection would be beneficial, the authors can discuss how the assay fairs to any antibody detection assay (IgM or IgG) or Nucleic acid-based detection if performed and emphasize early and late-stage detection.

Response: We think that 1.6 x 10^4/mL of the virus is just around the LOD since 2.5 x 10^4/mL was the LOD value in the additional experiment using NHP serum spike with the virus (See below). Regarding the second point, we modified discussion as follows: “While nucleic acid detection assays are currently used for early detection of MARV infection, they require specialized laboratories and personnel. Antibody detection serological tests are not ideal for early diagnosis as IgM antibodies can only be detected several days after the onset of symptoms, while IgG antibodies can be detected in the second week of infection and persist for years [35]” (Lines 196-200).

- How does this assay compare to any similar assays available?

Response: Thank you for the comments. As far as we know, there is no IC device available for Marburg and Ravn viruses. But it would be interesting if the comparison can be done in future studies.

- The LOD is determined based on the cell culture supernatant. Could the authors have used any alternate methods alongside? Also, any spiking in serum and whole blood samples with virus supernatant have been tried?

Response: Thank you for the suggestion. We tested NHP serum spiked with 2-fold dilutions of the virus and determined more accurate LOD value for the virus in NHP sera (2.5 x 10^4/mL). These data have been added in Table 4 in the revised manuscript (Lines 169-174).

- Mention the limitations of the study in the discussion.

Response: Thank you for the comment. We added the following paragraph in the last paragraph of Discussion.

“One limitation of the study is that a positive IC assay test could not be related to the onset of viremia due to the 3-day intervals between collection of the infected NHP serum samples, unlike our previous study on QuickNaviTM-Ebola where the collection was at shorter intervals and the majority of NHPs tested positive on onset of viremia (Yoshida et al., 2016). Another limitation is that there was a limited number of NHP infectious samples tested and that the IC assay was not tested for known positive and negative human samples” (Lines 248-254).

Reviewer 2 Report

Comments and Suggestions for Authors

In this manuscript the authors describe the development of a quick and easy-to-use diagnostic tool for MARV and RAVV viruses, which will be a  valuable asset for outbreak responses in Africa.

The experimental process is well described and easy to follow, and the results are straight-forward and clearly presented.

One addition that would be useful to include is how the test performs with blood samples compared to serum samples, since you indicate that either can be used.

It is understandable that animal samples are limited for testing of the LOD.  However, could uninfected blood or serum be spiked with different concentrations of stock viruses instead?  This could increase the "n" used to determine the LOD, provide a wider breadth of concentrations to be tested, and provide a more accurate LOD value. 

Similarly, has it ever been tested whether the species that the serum  originates from affect the LOD? Uninfected serum from NHPs and humans could potentially be compared with different concentrations of spiked virus.

In table 3 could the virus species be specified?  The only positive sample that wasn't detected by the IC assay was from MARV #5; if this was Ozolin this would be congruent with the LOD, however, the others (especially Musoke and Angola) should show up.

You indicate that it is not likely possible to create a single cross-reactive antibody against both MARV and EBOV viruses; however, are cocktails possible to use in a strip test, e.g. similar to the Flu + SARS2 test?  Or could a two antibodies potentially be fused and expressed as a single antibody?

Author Response

Reviewer #2

- In this manuscript the authors describe the development of a quick and easy-to-use diagnostic tool for MARV and RAVV viruses, which will be a valuable asset for outbreak responses in Africa.

The experimental process is well described and easy to follow, and the results are straight-forward and clearly presented.

Response: Thank you for your supportive comments.

- One addition that would be useful to include is how the test performs with blood samples compared to serum samples, since you indicate that either can be used.

Response: Thank you for the comments. Unfortunately, we do not have available whole blood samples of infected NHPs. However, we assume that there is no significant difference between serum and blood samples in the performance of our IC device since the same filter pad is used as our previously established RDT, QuickNavi-Ebola (Refs 17,18,45,46). When clinical whole blood samples were tested, this QuickNavi-Ebola showed comparable sensitivity to that obtained using serum samples of experimentally infected NHPs.

- It is understandable that animal samples are limited for testing of the LOD.  However, could uninfected blood or serum be spiked with different concentrations of stock viruses instead?  This could increase the "n" used to determine the LOD, provide a wider breadth of concentrations to be tested, and provide a more accurate LOD value.

Response: According to the suggestion. we tested NHP serum spiked with 2-fold dilutions of the virus and determined more accurate LOD value. These data have been added in Table 4 in the revised manuscript (Lines 69-174).

- Similarly, has it ever been tested whether the species that the serum originates from affect the LOD? Uninfected serum from NHPs and humans could potentially be compared with different concentrations of spiked virus.

Response: Thank you for the comments. It would be interesting if there is a difference among primate species in the sensitivity of the RDT. We have never tried to test this potential issue. However, we believe that it is beyond the scope of the present study to find a difference in LOD among animal species. Instead, we only tested several control NHP and human sera (data not shown) and found no nonspecific reaction in the IC assay.

- In table 3 could the virus species be specified? The only positive sample that wasn't detected by the IC assay was from MARV #5; if this was Ozolin this would be congruent with the LOD, however, the others (especially Musoke and Angola) should show up.

Response: Thank you for the comments. We added the variant name (Angola)(line 160). We think that 1.6 x 10^4/mL of the virus is just around the LOD since 1.0 x 10^4/mL of the virus could only be detected with single + in the tissue culture supernatant. It might not be surprising even if the IC assay could not detect 1.6 x 10^4/mL of the virus in the serum. Indeed, 2.5 x 10^4/mL was the LOD value in the additional experiment using NHP serum spike with the virus (See Table 4).

- You indicate that it is not likely possible to create a single cross-reactive antibody against both MARV and EBOV viruses; however, are cocktails possible to use in a strip test, e.g. similar to the Flu + SARS2 test?  Or could a two antibodies potentially be fused and expressed as a single antibody?

Response: Thank you for the comments. Yes, such antibody cocktails or recombinant antibodies may be used for a single line in a IC device. But, we are currently trying make a kit with 2 lines for Ebola and Marburg using different antibodies for each virus. It may be similar to Flu + SARS2 tests. To indicate the possibility of the use of antibody cocktails, we modified the last sentence of the third paragraph in Discussion as follows: “Thus, it might be difficult to produce highly cross-reactive mAbs to both orthoebolavirus and orthomarburgvirus, suggesting the use of antibody cocktails for developing pan-filovirus RDTs” (Lines 227-229).

Round 2

Reviewer 2 Report

Comments and Suggestions for Authors

Thank you for adding a few experiments and data.